# Comparative Effectiveness and Safety of Fractional Laser and Fractional Radiofrequency for Atrophic Acne Scars: A Retrospective Propensity Score Analysis

**DOI:** 10.3390/life15091379

**Published:** 2025-09-01

**Authors:** Chadakan Yan, Phichayut Phinyo, Yuri Yogya, Mati Chuamanochan, Rungsima Wanitphakdeedecha

**Affiliations:** 1Center for Clinical Epidemiology and Clinical Statistics, Faculty of Medicine, Chiang Mai University, 110 Intawaroros Road, Si Phum, Muang, Chiang Mai 50200, Thailand; chadakanyan4090@gmail.com; 2Department of Biomedical Informatics and Clinical Epidemiology (BioCE), Faculty of Medicine, Chiang Mai University, 110 Intawaroros Road, Si Phum, Muang, Chiang Mai 50200, Thailand; 3Musculoskeletal Science and Translational Research (MSTR) Center, Chiang Mai University, 110 Intawaroros Road, Si Phum, Muang, Chiang Mai 50200, Thailand; 4Department of Dermatology and Venereology, Universitas Padjadjaran, Bandung 40161, Indonesia; yuri.yogya@yahoo.com; 5Division of Dermatology, Department of Internal Medicine, Faculty of Medicine, Chiang Mai University, 110 Intawaroros Road, Si Phum, Muang, Chiang Mai 50200, Thailand; drmch117@gmail.com; 6Department of Dermatology, Faculty of Medicine Siriraj Hospital, Mahidol University, 2 Pran-nok Road, Bangkoknoi, Bangkok 10700, Thailand

**Keywords:** atrophic acne scar, fractional laser, fractional radiofrequency, energy-based device, Asian skin, clinical effectiveness

## Abstract

Fractional laser (FL) and fractional radiofrequency (FRF) are effective treatments for atrophic acne scars, yet comparative data in Asian populations with darker skin types remain limited. This retrospective cohort study compared the clinical effectiveness and safety of FL and FRF in Thai patients aged 18–60 years with Fitzpatrick skin types III–IV who underwent at least two treatment sessions between 2012 and 2023. Baseline characteristics were balanced using propensity score stratification, and missing data were addressed through multiple imputation with chained equations. The primary endpoint was the proportion of patients achieving ≥25% improvement in scarring at 6 months, with equivalence testing performed using a 20% margin. A total of 397 patients (254 FL, 143 FRF) were included, with balanced baseline characteristics after stratification. At 6 months, 88.1% of FRF-treated and 71.9% of FL-treated patients achieved the primary endpoint. FRF showed numerically greater mean improvement at all time points, though differences were not statistically significant. FL met the non-inferiority criterion but not equivalence. FRF was associated with significantly higher pain scores (*p* < 0.001), while adverse events, including post-inflammatory hyperpigmentation, were rare and similar between groups. Both modalities demonstrated meaningful clinical benefit and acceptable safety, although statistical equivalence could not be established and FRF was associated with greater procedural discomfort.

## 1. Introduction

Acne scarring is a prevalent and often persistent sequela of acne vulgaris, affecting up to 95% of individuals, with approximately 30% developing severe forms [1,2]. These scars result from an aberrant wound-healing response triggered by dermal inflammation, leading to disruption of collagen homeostasis [3]. Acne scars are broadly categorized as either atrophic—due to collagen loss—or hypertrophic/keloid, which result from excessive collagen deposition [4]. Among these, atrophic scars are most common and include ice pick (60–70%), boxcar (20–30%), and rolling scars (15–25%) [4]. In addition to their physical appearance, acne scars can significantly impact psychosocial well-being, contributing to low self-esteem, and impaired quality of life [5].

Treatment approaches vary depending on the scar type, severity, and patient skin characteristics. Modalities include surgical interventions such as subcision, punch excision, and elliptical excision, as well as non-surgical options like chemical peels, microneedling, dermabrasion, injectable fillers, lasers, and energy-based devices (EBDs) [6]. Over the past decade, EBDs—particularly lasers and radiofrequency (RF) technologies—have become a mainstay in scar management [7]. These devices promote dermal remodeling by delivering thermal energy to stimulate fibroblast activity and enhance collagen and elastin production, improving skin texture and tone [6].

Among lasers, ablative fractional lasers (AFLs), such as CO_2_ and Er:YAG, are highly effective for moderate-to-severe atrophic scars, producing clinical improvements ranging from 51% to 70% [8,9]. However, these lasers are associated with significant side effects, including erythema, discomfort, prolonged recovery, and post-inflammatory hyperpigmentation (PIH)—particularly in Asian skin [10,11]. A Cochrane review concluded that fractional lasers are comparable in efficacy to fractional RF and superior to non-fractional, non-ablative lasers [12]. While AFLs remain the preferred choice for deeper scars, less invasive RF modalities may be suitable for patients with mild-to-moderate scars or those seeking shorter downtime [8,9].

Fractional RF technology is categorized into two categories. One of them is a radiofrequency system utilizing a microneedle [13]. A newer generation of this technology could deliver bipolar RF energy via insulated or non-insulated microneedles directly into the dermis [14]. This technique produces controlled thermal damage into a deeper dermis while sparing the epidermis, allowing for deeper collagen remodeling with minimal surface injury [14]. Compared to FL, FRF has a lower risk of PIH, erythema, and crusting, making it particularly advantageous for patients with Fitzpatrick skin types III–VI [8,12]. Moreover, FRF does not rely on chromophore absorption, allowing safer use across all skin types [8]. Studies have shown that FRF provides scar improvement comparable to fractional lasers while offering enhanced dermal penetration, sustained neocollagenesis, and fewer adverse effects [12,15].

Despite their growing use, comparative evidence on the clinical effectiveness and safety of FL versus FRF in Asian populations remains limited. In particular, data from Thai patients with Fitzpatrick skin types III–IV are lacking. Given the heightened risk of PIH in darker skin tones, careful evaluation of treatment outcomes is essential. This study aims to compare the clinical effectiveness and safety profiles of fractional laser and fractional radiofrequency in Thai patients with atrophic acne scars. We hypothesized that fractional laser would be non-inferior to fractional radiofrequency in improving atrophic acne scars in Thai patients with Fitzpatrick skin types III–IV. The findings may inform clinical decision-making and support tailored treatment strategies for patients with darker skin tone.

## 2. Materials and Methods

A therapeutic efficacy research using a retrospective cohort design was conducted. The medical records of patients with atrophic acne scars at Siriraj Skin and Laser Center (SISL), Siriraj Hospital, Mahidol University, between 2012 and 2023 were collected and classified into fractional laser treatment and fractional radiofrequency treatment, according to the treatment modality chosen. Sample size was calculated based on prior data reporting that approximately 70% of patients treated with ablative fractional CO_2_ laser achieved greater than 25% improvement in scarring [16]. Assuming that the fractional radiofrequency group would have a similar response rate, an equivalence margin of 20% was prespecified as the maximum clinically acceptable difference. With a two-sided α of 0.05 and 80% power, a minimum of 111 patients per group (222 total) was required. This study was approved by the Siriraj Institutional Review Board (Si 735/2023) and conducted in accordance with the principles of the Declaration of Helsinki (1964) and its subsequent amendments. This study is reported in accordance with the Strengthening the Reporting of Observational Studies in Epidemiology (STROBE) guidelines for cohort studies (Appendix A) [17].

Patients aged 18–60 years diagnosed with facial atrophic acne scars (icepick, boxcar, or rolling) who underwent at least two sessions of either fractional laser or fractional radiofrequency at Siriraj Hospital between 2012 and 2023 were included. Patients were excluded if baseline or post-treatment clinical photographs were unavailable. The two-session requirement was based on previous findings indicating the clinical significance [10]. The primary objective was to compare the proportion of those who achieved ≥25% clinical improvement grading by two independent dermatologists between fractional laser and fractional radiofrequency treatments using photographic assessment. Secondary objectives included comparing the percentage of clinical improvement grading by two independent dermatologists of both modalities and adverse events.

Descriptive statistics were used to summarize baseline characteristics. Continuous variables were reported as mean ± standard deviation (SD) or median (interquartile range, IQR), and compared using independent t-tests or Wilcoxon rank-sum tests as appropriate. Categorical variables were presented as frequencies and percentages and compared using Fisher’s exact test. To address baseline confounding, confounding-by-indication/contraindication, due to non-randomized treatment assignment, propensity score (PS) stratification was performed based on pre-treatment covariates, including age, sex, scar type, scar severity, scar duration, and skin type. Covariate balance before and after stratification was assessed using standardized differences (STD).

Missing data for baseline characteristics were handled using k-nearest neighbor (kNN) imputation, while missing longitudinal outcomes were addressed with multiple imputation by chained equations using predictive mean matching. For the primary outcome, the proportion of patients with ≥25% improvement was analyzed using linear mixed-effects models with random intercepts. Equivalence testing was conducted with a predefined margin of 20%, applying two one-sided t-tests (TOST) on the proportion of patients achieving ≥25% improvement at 6 months. The percentage of clinical improvement at 1, 3, and 6 months was compared between groups using marginal effects post-estimation from the linear mixed-effects models. All analyses were performed using Stata version 17 (StataCorp, College Station, TX, USA). A two-sided *p*-value < 0.05 was considered statistically significant.

## 3. Results

### 3.1. Baseline Characteristics and Propensity Score Stratification

A total of 397 patients with atrophic acne scars were included in the study: 254 received FL treatment and 143 received FRF treatment. Before PS stratification, substantial differences were observed in baseline characteristics, particularly in scar duration (15.30 ± 7.07 vs. 8.47 ± 3.96 years; STD = −1.040) and acne scar type (STD = −0.139). Following PS stratification, all variables achieved acceptable balance, with STDs reduced to <0.1, indicating that the FL and FRF groups were well-balanced on baseline covariates (Table 1, Appendix A). The original dataset prior to imputation is shown in Appendix A, and treatment device parameters are summarized in Appendix A. The study flow diagram is presented in Appendix A.

### 3.2. Efficacy Outcomes

At 1-, 3-, and 6-month follow-ups, the proportion of patients achieving ≥25% clinical improvement was higher in the FRF group compared to the FL group: 73% vs. 60% (*p* = 0.080), 82% vs. 69% (*p* = 0.072), and 88% vs. 72% (*p* = 0.031), respectively. The mean differences were 13% at 1 and 3 months (95% CI: −0.02 to 0.28 and −0.01 to 0.29), and 16% at 6 months (95% CI: 0.01 to 0.31) (Figure 1, Table 2). Similarly, the mean percentage of improvement also numerically favored FRF at all time points: 41.68% vs. 37.98% at 1 month (difference: 3.70%, 95% CI: −4.01 to 11.41; *p* = 0.347), 48.71% vs. 43.27% at 3 months (difference: 5.43%, 95% CI: −2.28 to 13.15; *p* = 0.167), and 54.89% vs. 47.53% at 6 months (difference: 7.35%, 95% CI: −0.29 to 15.00; *p* = 0.059) (Figure 2, Table 2). Patients in the FRF group received more treatment sessions on average than those in the FL group (3.41 ± 0.49 vs. 3.21 ± 0.62).

Equivalence testing was performed using the TOST method with a predefined margin of ±20% for the proportion of patients achieving ≥25% improvement at 6 months. While the point estimate for fractional radiofrequency (FRF) was numerically higher than fractional laser (FL), the 95% confidence interval crossed both the null and equivalence margins, indicating that statistical equivalence could not be concluded. The *p*-value for the upper bound of the equivalence test was not statistically significant (*p* = 0.306). However, the lower bound remained above the non-inferiority threshold, supporting that FL is not inferior to FRF (Figure 3).

### 3.3. Safety and Tolerability

Patients receiving FRF reported significantly higher pain scores (5.65 ± 1.74) than those treated with FL (4.14 ± 1.83), *p* < 0.001; STD = −0.846 (Table 3). Other adverse events were uncommon and comparable between groups. Post-inflammatory hyperpigmentation was observed in 5.42% of FL-treated patients and 4.71% of FRF-treated patients (*p* = 1.000). Acneiform eruptions occurred in 6.67% of FL patients and 3.53% of FRF patients (*p* = 0.421), and post-treatment scarring occurred in 0.83% and 3.03%, respectively (*p* = 0.191). No infections were reported in either group.

## 4. Discussion

A propensity score–adjusted retrospective cohort study was done to compare the safety and effectiveness of fractional laser and fractional radiofrequency in treating atrophic acne scars in Fitzpatrick skin types III–IV. Both energy-based device modalities demonstrated progressive clinical improvement over time. Although outcomes numerically favored FRF at 1-, 3-, and 6-month follow-ups, these differences were not statistically significant.

Using a predefined equivalence margin of 20%, we found that although the confidence intervals for the differences in response rates crossed both the null and equivalence thresholds—precluding a definitive conclusion of superiority or equivalence—the lower bound remained above the non-inferiority threshold. This supports the conclusion that FL is not inferior to FRF in terms of achieving ≥25% clinical improvement. It should be noted that at the 6-month follow-up, FRF demonstrated a statistically higher proportion of patients achieving ≥25% improvement. However, equivalence was not established using the TOST approach, and only non-inferiority of FL relative to FRF could be confirmed. These findings highlight the importance of distinguishing statistical significance from clinical significance: while FRF showed a numerical and statistically significant advantage at 6 months, both modalities provided clinically meaningful improvements. Thus, the results should be interpreted cautiously, and further randomized controlled studies are warranted to validate these findings.

This study incorporated various fractional laser modalities, including fractional CO_2_, Er:YAG, and picosecond lasers, each with distinct efficacy and safety profiles. Unlike ablative CO_2_ and Er:YAG lasers, fractional picosecond lasers achieve dermal remodeling primarily through laser-induced optical breakdown and photoacoustic effects rather than photothermal ablation [18]. This mechanistic distinction may account for their generally more modest clinical outcomes but improved tolerability, and their inclusion within the FL category may have attenuated the overall efficacy observed in our analysis. Fractional radiofrequency devices are also heterogeneous, comprising microneedle systems that deliver bipolar energy through dermal penetration and sublative platforms that apply energy superficially via electrode arrays [19]. These differences in tissue interaction may influence treatment response and should be taken into account when interpreting our findings. The heterogeneity within the FL group similarly represents an important limitation, as each modality has unique mechanisms, penetration depths, and clinical outcomes. In this study, fractional lasers were analyzed collectively to reflect their shared principle of fractional photothermolysis; however, subgroup analyses were not feasible given the limited sample size and the risk of reduced statistical power. Future studies with larger cohorts are warranted to allow more refined comparisons between individual platforms. Moreover, Ablative lasers such as CO_2_ and Er:YAG are recognized for their superior clinical efficacy in treating atrophic acne scars but are often associated with increased pain, erythema, and prolonged recovery time [20]. Conversely, picosecond lasers are generally better tolerated and carry a lower risk of post-inflammatory hyperpigmentation, rendering them more suitable for patients with darker skin types, although they typically provide more modest clinical outcomes [21]. The numerically superior improvement observed in the FRF group in our study may, in part, be attributed to variability in the efficacy of the fractional laser modalities employed, particularly the inclusion of a fractional picosecond laser. Experts’ consensus acknowledges that ablative fractional laser therapies, including CO_2_ and Er:YAG lasers, are the most efficacious approach for addressing atrophic acne scars [8]. In accordance with previous research [22], the CO_2_ FL generally yields superior therapeutic results compared to the fractional picosecond laser, particularly with an increased number of treatment sessions [23].

Our findings are consistent with prior comparative studies. Hendel et al. conducted a randomized split-face trial demonstrating that both fractional CO_2_ laser and FRF yielded significant scar improvement, but FRF was associated with greater tolerability, less erythema, and more rapid recovery [24]. Similarly, Rajput et al. reported equivalent clinical efficacy but found FRF to have a more favorable safety profile and shorter downtime [25]. Sriram et al. also observed comparable scar improvement, with a lower incidence of pigmentary complications and adverse events in the FRF group, particularly in Fitzpatrick skin types III–IV [26]. Both methods were safe, efficient, and had tolerable adverse effects. Therefore, these two techniques can effectively replace one another, and if required, their combination can yield superior outcomes in the treatment of atrophic scars [27]. In contrast, recent network meta-analyses reported that ablative fractional lasers provide greater scar improvement than non-ablative fractional lasers and radiofrequency modalities, albeit with more discomfort and longer recovery. Li et al. (2023) found ablative fractional laser significantly more effective than non-ablative fractional laser and radiofrequency, while Wang et al. (2023) ranked fractional CO_2_ and Er:YAG lasers highest in dermatologist-rated improvement but also associated them with higher pain and PIH risk [28,29]. These findings differ from our real-world data, where FL and FRF showed comparable outcomes with low adverse event rates. The discrepancy likely reflects methodological differences: NMAs integrate heterogeneous trials across diverse populations and devices, whereas our study captures routine practice in Thai patients with Fitzpatrick skin types III–IV.

In our study, both modalities were well tolerated, with adverse events being infrequent and self-limited. The most notable difference was greater procedural discomfort associated with FRF, likely due to deeper dermal stimulation. Although pain scores were statistically higher in the FRF group (*p* < 0.001), they remained within a clinically acceptable range, and no patients discontinued treatment due to discomfort. Mild-to-moderate procedural pain is generally considered tolerable in dermatologic practice and rarely compromises adherence. Furthermore, while adverse events such as PIH, acneiform eruptions, and post-treatment scarring were uncommon and comparable between groups, our study was not powered to reliably detect very rare complications such as infection or persistent scarring. This limitation should be taken into account when interpreting the safety profile of these modalities. The incidence of PIH was low in both groups (5.4% for FL vs. 4.7% for FRF), but lower than that reported by Sriram et al., who found PIH rates of 13.3% for FL and 3.3% for FRF [26]. This discrepancy may be attributable to differences in device parameters, post-procedural care, or operator experience. Erythema and epidermal crusting—typical of ablative laser resurfacing—were also more pronounced in the FL group, consistent with the findings of Hendel et al. [24]. These reactions are expected due to the epidermal disruption inherent to ablative laser delivery, whereas FRF has minimal injury to the epidermis, reducing visible downtime and post-treatment morbidity. In our study, acneiform eruptions and post-treatment scarring were rare and not significantly different between groups. A slightly higher incidence of acneiform eruptions was observed in the FL group, although this difference did not reach statistical significance. The mechanisms underlying these eruptions are not fully elucidated, but proposed explanations include transient follicular occlusion, inflammatory responses to thermal injury, or disruption of the cutaneous microbiome and barrier function. Patients with acne-prone skin may be particularly susceptible, as previously reported in a large review of over 700 patients undergoing fractional laser treatments [30]. Importantly, these eruptions are typically self-limited and manageable with standard acne therapies. No cases of infection or delayed healing were observed in either arm, reinforcing the procedural safety of both modalities when performed under appropriate conditions.

These outcomes align with the mechanisms of dermal remodeling. FL induces controlled microthermal injury in the superficial dermis, promoting neocollagenesis and re-epithelialization [31]. In contrast, FRF delivers thermal energy deeper into the dermis while sparing the epidermis, enabling effective remodeling with reduced PIH risk and downtime—beneficial for darker skin types [32]. Taken together, these findings support FL and FRF as effective and safe options for the management of atrophic acne scarring in Asian populations. The reduced incidence of pigmentary adverse events and shorter recovery time associated with FRF may be particularly advantageous for patients with higher melanin content or those seeking minimal downtime. Nonetheless, the greater procedural discomfort with FRF warrants consideration during patient consultation. Ultimately, treatment selection should be individualized based on clinical and patient-centered factors.

This study has several limitations inherent to its retrospective design. Although propensity score stratification was applied to reduce baseline confounding, residual bias cannot be excluded. Treatment allocation was based on clinical judgment rather than randomization, and factors such as physician preference, operator experience, patient expectations, and device parameter selection may have influenced outcomes. Additionally, adherence to post-procedural care was not captured, which may have contributed to variability in treatment response.

Another limitation is the heterogeneity of devices included. The FL group included CO_2_, Er:YAG, and picosecond lasers, which differ substantially in mechanism, depth of penetration, and expected outcomes, while the FRF group included both microneedle-based and sublative systems with distinct tissue interactions. Although this reflects real-world practice, pooling across heterogeneous platforms may have introduced variability and limits the generalizability of our results. Clinical improvement was assessed through photographic grading by two dermatologists, with the mean score used to minimize subjectivity; however, inter-rater reliability (e.g., Kappa statistic) was not formally evaluated. Finally, while our total sample size exceeded the prespecified requirement, the unequal distribution between groups (FL = 254, FRF = 143) may have reduced power for between-group and subgroup analyses. The single-center design and predominance of Fitzpatrick skin types III–IV further limit generalizability. These limitations underscore the need for cautious interpretation of our findings and support the need for prospective randomized controlled trials in larger and more diverse populations.

## 5. Conclusions

This propensity score-adjusted retrospective cohort study demonstrates that both fractional laser and fractional radiofrequency are effective and well-tolerated energy-based devices for the treatment of atrophic acne scars in Thai patients with Fitzpatrick skin types III–IV. While FRF showed a numerical trend toward greater clinical improvement at 1-, 3-, and 6-month follow-up intervals, these differences did not consistently achieve statistical significance. Both modalities exhibited low rates of adverse events, with FRF associated with greater procedural pain. Given these findings, both FL and FRF remain viable options in clinical practice, and treatment selection should be individualized based on scar characteristics, patient tolerance, and expectations regarding recovery time and comfort.

## Figures and Tables

**Figure 1 life-15-01379-f001:**
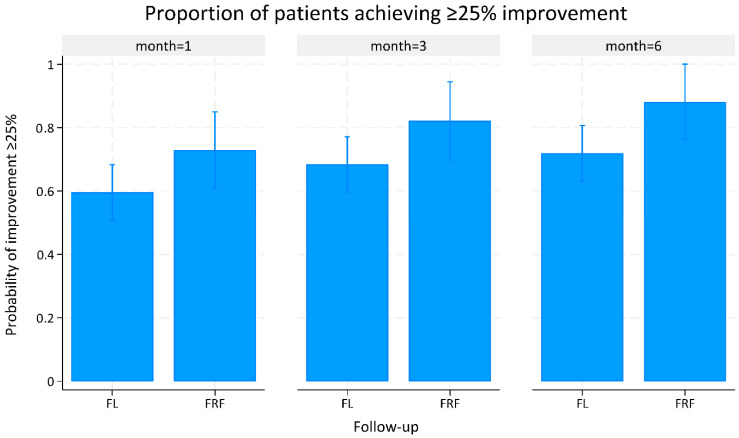
Proportion of patients achieving ≥25% clinical improvement at 1, 3, and 6 months after treatment with fractional laser versus fractional radiofrequency.

**Figure 2 life-15-01379-f002:**
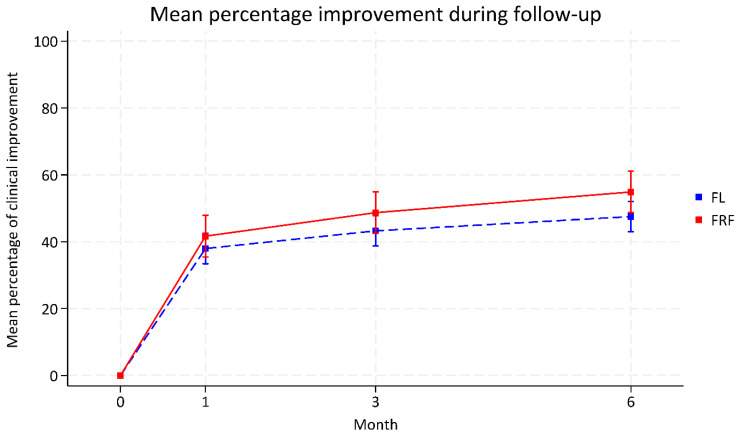
Mean percentage of clinical improvement following fractional laser and fractional radiofrequency treatment at 1-, 3-, and 6-month follow-up visits.

**Figure 3 life-15-01379-f003:**
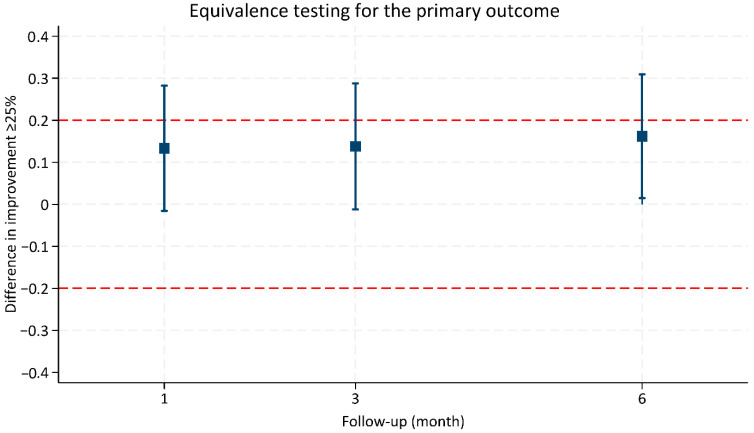
Equivalence testing of fractional laser versus fractional radiofrequency for achieving ≥25% clinical improvement at 6 months.

**Table 1 life-15-01379-t001:** Demographic and Clinical Characteristics of Patients with Acne Scarring: Comparison of Fractional Laser and Fractional Radiofrequency Treatment Groups Before and After Propensity Score Stratification.

Characteristics	Fractional Laser Treatment (n = 254)	Fractional Radiofrequency Treatment (n = 143)	Standardized Difference (STD) Before PS Stratification	Standardized Difference (STD) After PS Stratification
	N (%) or Mean ± SD	N (%) or Mean ± SD		
Demographic factors				
Age (years)	33.17 ± 7.41	34.29 ± 10.67	**0.216**	0.051
Gender			−0.079	−0.080
Male	96 (37.80)	42 (29.37)		
Female	158 (62.20)	101 (70.63)		
Skin phototypes				
III	56 (22.05)	37 (25.87)	**0.110**	−0.017
IV	166 (65.35)	90 (62.94)	**−0.115**	0.005
V	32 (12.60)	16 (11.19)	0.026	0.016
Clinical factors				
Scar age (years)	15.30 ± 7.07	8.47 ± 3.96	**−1.040**	−0.045
Severity of acne scar			0.092	−0.020
Almost clear—Mild	27 (10.63)	7 (4.90)		
Moderate—very severe	227 (89.37)	136 (95.10)		
Type of acne scar			**−0.139**	0.051
1	54 (21.26)	44 (30.77)		
>1	200 (78.74)	99 (69.23)		

SD = standard deviation; STD = standardized difference. Significant between-group differences (absolute standardized difference > 0.1) are presented in bold.

**Table 2 life-15-01379-t002:** Participant Clinical Improvement Outcomes at 1-, 3-, and 6-Month Follow-Up.

Outcome	Fractional Laser Treatment (95% CI)	Fractional Radiofrequency Treatment (95% CI)	Mean Difference (95% CI)	*p* Value
Proportion of ≥25% clinical improvement				
1-month follow-up	0.60 (0.51 to 0.68)	0.73 (0.61 to 0.85)	0.13 (−0.02 to 0.28)	0.080
3-month follow-up	0.69 (0.60 to 0.77)	0.82 (0.70 to 0.94)	0.13 (−0.01 to 0.29)	0.072
6-month follow-up	0.72 (0.63 to 0.81)	0.88 (0.76 to 1.00)	0.16 (0.01 to 0.31)	**0.031** *
Percentage of clinical improvement				
1-month follow-up	37.98 (33.44 to 42.52)	41.68 (35.44 to 47.92)	3.70 (−4.01 to 11.41)	0.347
3-month follow-up	43.27 (38.73 to 47.81)	48.71 (42.47 to 54.94)	5.43 −2.28 to 13.15	0.167
6-month follow-up	47.53 (43.00 to 52.05)	54.89 (48.70 to 61.07)	7.35 −0.29 to 15.00)	0.059

Outcomes were imputed using multiple imputation with chained equations. * *p*-value for equivalence was not statistically significant for the upper equivalence margin (*p* = 0.306).

**Table 3 life-15-01379-t003:** Safety Outcomes of Fractional Laser and Fractional Radiofrequency Treatments.

Side Effects/Adverse Events	Fractional Laser Treatment (n = 254)	Fractional Radiofrequency Treatment (n = 143)	*p*-Value	Standardized Difference (STD)
Average Pain score (n = 203)	4.14 ± 1.83	5.65 ± 1.74	**<0.001**	**−0.846**
Post-inflammatory hyperpigmentation (n = 325)	13 (5.42)	4 (4.71)	1.000	0.032
Acneiform eruption (n = 325)	16 (6.67)	3 (3.53)	0.421	**0.143**
Infection (n = 325)	0 (0)	0 (0)	-	-
Post-treatment scar (n = 372)	2 (0.83)	4 (3.03)	0.191	**−0.160**

Outcomes were not imputed; data are reported only for patients with complete safety information. Statistically significant *p*-value and significant between-group differences (absolute standardized difference > 0.1) are presented in bold.

## Data Availability

The datasets generated and analyzed during the current study are available from the corresponding author upon reasonable request.

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
