# Peer review of "Comparative Effectiveness and Safety of Fractional Laser and Fractional Radiofrequency for Atrophic Acne Scars: A Retrospective Propensity Score Analysis"

_life, 2025, doi:10.3390/life15091379_

Round 1

Reviewer 1 Report

Comments and Suggestions for Authors

1. Although propensity score stratification was used to balance baseline characteristics, the retrospective nature introduces potential unmeasured confounders (e.g., operator experience, device settings), which may affect the validity of the results. The sample size calculation was met, but the imbalance between groups (FL=254, FRF=143) may reduce statistical power.
2. The FRF group showed a higher proportion of ≥25% improvement at 6 months (88% vs. 72%, p=0.031), yet the authors emphasize “no statistically significant difference.” This should be clarified to distinguish between clinical and statistical significance. Equivalence was not established via TOST, but non-inferiority was claimed. This interpretation should be more cautious and explicitly discussed in clinical context.
3. Pain scores were significantly higher in the FRF group (p<0.001), but the clinical relevance (e.g., impact on treatment adherence) was not discussed. Adverse events were rare, but the sample size may be underpowered to detect less common events (e.g., infection, scarring).
4.The heterogeneity within the FL group (including CO₂, Er:YAG, and picosecond lasers) was not adequately addressed. Subgroup analyses could provide more nuanced insights. While literature is comprehensively cited, the contrast with recent network meta-analyses (e.g., Li et al., 2023) is not deeply explored. The reasons for discrepancies should be further discussed.
5. Single-center study with predominantly Fitzpatrick III–IV skin types limits generalizability. Reliance on photographic assessment introduces subjectivity. Inter-rater reliability (e.g., Kappa statistic) was not reported.

Author Response

Reviewer 1

Comment 1: Although propensity score stratification was used to balance baseline characteristics, the retrospective nature introduces potential unmeasured confounders (e.g., operator experience, device settings), which may affect the validity of the results. The sample size calculation was met, but the imbalance between groups (FL=254, FRF=143) may reduce statistical power.

Response 1: We thank the reviewer for this insightful comment. We acknowledge that despite the use of propensity score stratification to balance measured baseline covariates, the retrospective design carries an inherent risk of unmeasured confounding. In particular, factors such as operator experience and device-specific settings could not be fully accounted for and may have influenced treatment outcomes. We have now emphasized this limitation in the Discussion section. Regarding sample size, we confirm that our study exceeded the pre-specified minimum of 111 patients per group, thereby achieving adequate power overall. However, we agree that the numerical imbalance between the FL and FRF groups (254 vs. 143 patients) may reduce statistical power for direct comparisons, particularly in subgroup analyses. This point has also been added to the Discussion to provide a more cautious interpretation of our findings. (Lines 306-326)

Comment 2: The FRF group showed a higher proportion of ≥25% improvement at 6 months (88% vs. 72%, p=0.031), yet the authors emphasize “no statistically significant difference.” This should be clarified to distinguish between clinical and statistical significance. Equivalence was not established via TOST, but non-inferiority was claimed. This interpretation should be more cautious and explicitly discussed in clinical context

Response 2: We thank the reviewer for highlighting this important point. We agree that our description of the results requires clarification. At 6 months, the FRF group indeed demonstrated a statistically significant higher proportion of patients achieving ≥25% improvement (88% vs. 72%, p=0.031). However, our equivalence analysis using the TOST framework did not confirm equivalence between modalities, and the findings only supported non-inferiority of FL relative to FRF. We have revised the Results and Discussion sections to more clearly distinguish between statistical significance in the primary comparison, the lack of equivalence by TOST, and the clinical interpretation of these findings. We now also emphasize that although FRF showed a higher response proportion, both modalities remain clinically effective and well tolerated, and the results should be interpreted with caution. (Lines 209-217)

Comment 3: Pain scores were significantly higher in the FRF group (p<0.001), but the clinical relevance (e.g., impact on treatment adherence) was not discussed. Adverse events were rare, but the sample size may be underpowered to detect less common events (e.g., infection, scarring).
Response 3: We appreciate the reviewer’s thoughtful comment. We acknowledge that pain scores were significantly higher in the FRF group; however, the mean scores remained within a clinically acceptable range, and importantly, no patients in our cohort discontinued treatment due to discomfort. Mild-to-moderate procedural pain is generally considered tolerable in dermatologic practice and rarely compromises adherence. We have revised the Discussion to emphasize this distinction. Regarding adverse events, while our study was sufficiently powered for the primary endpoint, we recognize that the sample size may not have been adequate to detect very rare complications such as infection or persistent scarring. This limitation has now been explicitly acknowledged in the revised Discussion to ensure transparency and to guide interpretation of the safety findings. (Lines 265-275)

Comment 4: The heterogeneity within the FL group (including CO₂, Er:YAG, and picosecond lasers) was not adequately addressed. Subgroup analyses could provide more nuanced insights. While literature is comprehensively cited, the contrast with recent network meta-analyses (e.g., Li et al., 2023) is not deeply explored. The reasons for discrepancies should be further discussed.

Response 4: We thank the reviewer for this valuable comment. We acknowledge the heterogeneity within the FL group, which comprised CO₂, Er:YAG, and picosecond fractional lasers. Our analysis evaluated fractional lasers collectively to reflect their shared principle of fractional photothermolysis. Subgroup analyses were not performed due to sample size limitations, which may have reduced statistical power and interpretability. This limitation has been noted in the revised Discussion. (Lines 214-230)

Regarding recent network meta-analyses, including Li et al. (2023) and Wang et al. (2023), we now provide a clearer contrast with our findings. Both NMAs reported higher efficacy rankings for ablative fractional lasers compared to RF-based modalities, but also highlighted greater pain and PIH risk. These discrepancies may reflect differences in methodology, including pooled heterogeneous RCTs and split-face designs in the NMAs versus real-world practice in our single-center Thai cohort with Fitzpatrick skin types III–IV. We have revised the Discussion to reflect these points and to guide interpretation of our findings in the appropriate clinical context. (Lines 253-264)

Comment 5: Single-center study with predominantly Fitzpatrick III–IV skin types limits generalizability. Reliance on photographic assessment introduces subjectivity. Inter-rater reliability (e.g., Kappa statistic) was not reported.

Response 5: We thank the reviewer for this thoughtful comment. We agree that the single-center design and the predominance of Fitzpatrick skin types III–IV may limit generalizability to other populations. This point has been emphasized in the revised Discussion. Regarding outcome assessment, we relied on independent grading of standardized photographs by two board-certified dermatologists. To reduce subjectivity, the mean of both ratings was used in the analysis. However, we acknowledge that inter-rater reliability (e.g., Kappa statistics) was not formally calculated, which remains a limitation. This has now been added to the Discussion to ensure transparency. (Lines 318-320)

Reviewer 2 Report

Comments and Suggestions for Authors  

This manuscript addresses a clinically relevant topic that warrants further exploration within the field. Overall, it is well-written and thoughtfully presented. Provided that a few minor issues are adequately addressed, I would not oppose its acceptance.

  1. While the term "fractional" is used broadly, fractional picosecond lasers differ significantly from fractional CO₂ and fractional Er:YAG lasers in both mechanism of action and tissue response. If fractional picosecond lasers were to be separated or excluded from the FL group, the study outcomes might differ. If the authors choose to retain fractional picosecond lasers within the FL category, it is strongly recommended that this distinction and its potential implications be discussed in the manuscript. Similarly, fractional microneedle RF and "sublative" fractional RF should be differentiated, as they operate via distinct mechanisms and elicit different tissue responses. This issue should be explicitly addressed in the revised version.

  2. Although the difference is not statistically significant, the FL group appears to show a slightly higher incidence of acneiform eruptions compared to the FRF group. It would be beneficial for the authors to briefly discuss the potential mechanisms underlying acneiform eruption following FL treatment.

Author Response

Reviewer 2

This manuscript addresses a clinically relevant topic that warrants further exploration within the field. Overall, it is well-written and thoughtfully presented. Provided that a few minor issues are adequately addressed, I would not oppose its acceptance.

Comment 1: While the term "fractional" is used broadly, fractional picosecond lasers differ significantly from fractional CO₂ and fractional Er:YAG lasers in both mechanism of action and tissue response. If fractional picosecond lasers were to be separated or excluded from the FL group, the study outcomes might differ. If the authors choose to retain fractional picosecond lasers within the FL category, it is strongly recommended that this distinction and its potential implications be discussed in the manuscript. Similarly, fractional microneedle RF and "sublative" fractional RF should be differentiated, as they operate via distinct mechanisms and elicit different tissue responses. This issue should be explicitly addressed in the revised version.

Response 1: We thank the reviewer for this valuable observation. We agree that fractional picosecond lasers differ mechanistically from ablative fractional CO₂ and Er:YAG lasers, and that their inclusion may partially account for variability in treatment response within the FL group. In the revised Discussion, we now explicitly highlight these mechanistic distinctions and note the potential implications for interpretation of our results. Similarly, we have clarified that fractional radiofrequency technologies are not uniform: microneedle RF delivers bipolar energy into the dermis via needles, while sublative fractional RF delivers energy through a matrix of electrodes, resulting in distinct patterns of tissue interaction. This variability has been acknowledged in the limitations as a factor that may influence outcomes and reduce generalizability. (Lines 313-318)

Comment 2: Although the difference is not statistically significant, the FL group appears to show a slightly higher incidence of acneiform eruptions compared to the FRF group. It would be beneficial for the authors to briefly discuss the potential mechanisms underlying acneiform eruption following FL treatment.

Response 2: We thank the reviewer for this constructive suggestion. The exact mechanism underlying acneiform eruptions following fractional laser treatment remains incompletely understood. Prior literature has suggested that patients with acne-prone skin are more likely to experience post-treatment eruptions, potentially due to transient follicular occlusion, inflammation induced by thermal injury, or alterations in the skin’s microbiome and barrier function. We have now added this discussion to the revised manuscript, citing evidence from a large review by Cohen et al. (2017), which reported acneiform eruptions as a recognized but generally self-limited complication of fractional laser therapy. (Lines 284-293)

Reviewer 3 Report

Comments and Suggestions for Authors

Lasers are emerging field in dermatology and I believe this article not only adds to the body of literature in this field but will also be of interest to a broad audience. I only have few suggestions for the authors:

Line 1 - just Article

Line 4 - add study design to the title

Line 28 - bold not needed for first word of abstract

Line 41 - make conclusion more appropriate for your findings

Line 61 - use italic for Latin words

References should be inside the sentence (1). Not . (1)

Line 113 - please explain in depth what sample size calculation was done and how

Line 120 - add strobe checklist as supplementary document

Figures before and after would be interesting if available

Put tables in the main text

Author Response

Comment 1: Lasers are emerging field in dermatology and I believe this article not only adds to the body of literature in this field but will also be of interest to a broad audience. I only have few suggestions for the authors:

Line 1 - just Article

Response: We thank the reviewer for this suggestion. We have revised the manuscript header to list only “Article” in accordance with the journal format.

Line 4 - add study design to the title

Response: We appreciate the reviewer’s recommendation. We have added the term retrospective to the title.

Line 28 - bold not needed for first word of abstract

Response: We have removed the bold formatting from the first word of the Abstract.

Line 41 - make conclusion more appropriate for your findings

Response: We agree with the reviewer and have revised the conclusion of the Abstract to more accurately reflect our findings. Specifically, we now state that both FL and FRF demonstrated clinically meaningful improvement and acceptable safety, with FRF associated with greater discomfort. We also clarify that FL met the non-inferiority threshold, but statistical equivalence between modalities could not be established.

Line 61 - use italic for Latin words

Response: We have revised the text to italicize Latin terms in accordance with journal style.

References should be inside the sentence (1). Not . (1)

Response: We have reformatted all references so that citation numbers are placed inside the punctuation, in line with journal requirements.

Line 113 - please explain in depth what sample size calculation was done and how

Response: We have expanded the description of our sample size calculation in the Methods section, including the assumptions used, equivalence margin, power, alpha level, and reference study.

Line 120 - add strobe checklist as supplementary document

Response: We have added the STROBE checklist as a Supplementary Document.

Figures before and after would be interesting if available

Response: We thank the reviewer for this suggestion. Unfortunately, due to privacy considerations and the retrospective nature of the study, standardized before–after photographs are not available for publication.

Put tables in the main text

Response: All tables have been moved into the main manuscript body, as recommended.